# Does 'online confidence' predict application success and later academic performance in medical school? A UK-based national cohort study

Paul A Tiffin [1,2] Lewis W Paton [1]

[1]Department of Health Sciences, University of York, York, UK
[2]Health Professions Education Unit, Hull York Medical School, Heslington, UK

**Correspondence to**
Dr Paul A Tiffin;
paul.tiffin@york.ac.uk

## ABSTRACT

**Objectives** The UK Clinical Aptitude Test (UKCAT) previously piloted an assessment of 'online confidence', where candidates were asked to indicate how confident they were with their answers. This study examines the relationship between these ratings, the odds of receiving an offer to study medicine and subsequent undergraduate academic performance.

**Design** National cohort study.

**Setting** UK undergraduate medical selection.

**Participants** 56 785 UKCAT candidates who sat the test between 2013 and 2016 and provided valid responses to the online confidence pilot study.

**Primary outcome measures** Two measures of 'online confidence' were derived: the well-established 'confidence bias', and; a novel 'confidence judgement' measure, developed using Item Response Theory in order to derive a more sophisticated metric of the ability to evaluate one's own performance on a task. Regression models investigated the relationships between these confidence measures, application success and academic performance.

**Results** Online confidence was inversely related to cognitive performance. Relative underconfidence was associated with increased odds of receiving an offer to study medicine. For 'confidence bias' this effect was independent of potential confounders (OR 1.48, 1.15 to 1.91, p=0.002). While 'confidence judgement' was also a univariable predictor of application success (OR 1.22, 1.01 to 1.47, p=0.04), it was not an independent predictor. 'Confidence bias', but not 'confidence judgement', predicted the odds of passing the first year of university at the first attempt, independently of cognitive performance, with relative underconfidence positively related to academic success (OR 3.24, 1.08 to 9.72, p=0.04). No non-linear effects were observed, suggesting no 'sweet spot' exists in relation to online confidence and the outcomes studied.

**Conclusions** Applicants who either appear underconfident, or are better at judging their own performance on a task, are more likely to receive an offer to study medicine. However, online confidence estimates had limited ability to predict subsequent academic achievement. Moreover, there are practical challenges to evaluating online confidence in high-stakes selection.

## Strengths and limitations of this study

► This study used a large, national data set.
► This is, to our knowledge, the first study to link 'online confidence' of medical school applicants to subsequent outcomes at medical school.
► This study introduces a novel, alternative approach to measuring 'online confidence', using Item Response Theory.
► The 'online confidence' tests were piloted in low-stakes conditions, and therefore it is unclear whether our results would generalise to high-stakes settings.
► It was not possible to link 'online confidence' with previously piloted 'self-report' confidence measures, and thus we could not compare confidence as a trait and as a metacognitive ability.

## INTRODUCTION

The safe and effective practice of medicine can be assumed to require an accurate appraisal of one's own abilities. Indeed, overconfidence can lead to diagnostic errors.[1] Thus, the prospect of being able to accurately estimate ability is attractive to medical selectors. Competition to study medicine is high; in the UK there are around 11 applications for every available place.[2] As such, aptitude tests are commonly used as part of the selection procedure, partly in an attempt to effectively discriminate between large numbers of similarly high-performing candidates. Such cognitive test scores are also intended to predict future academic achievement in medical education, and there is accumulating evidence that they do this to some extent.[3–5]

There is an increased recognition of the essential role that 'non-academic' attributes play in a successful medical career. Indeed, most issues associated with professional malpractice are related to the personality and interpersonal functioning of the doctor involved, rather than a lack of clinical knowledge or skills.[6] Consequently there has been a

1. What is the best interpretation of the following coded message: 2, 11, 16, C, H, 9, 4

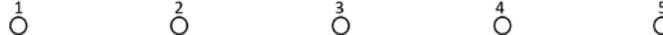

- ☐ A) Today we are chatting that the risk of house fires
- ☐ B) People talk too much about house fires
- ☐ C) We had a nice chat around the fires
- ☐ D) People talk too much on domestic disasters
- ☐ E) People are talking about yesterday's house fire

How confident are you that the answer you gave was right?

1 2 3 4 5
○ ○ ○ ○ ○

**Figure 1** An example of an item from the decision analysis section of the UK Clinical Aptitude Test within the confidence rating pilot.

growing drive to evaluate 'personal qualities', aside from intellectual ability and academic achievement, as part of medical selection at different career stages.[7–9] However, measuring such traits in a high-stakes selection situation poses a number of practical challenges.[10]

In a wider educational context, the relationship between 'self-beliefs' and academic performance have been well researched, from the late 1970s.[11 12] These 'self-beliefs' predict academic achievement to varying degrees. A literature review reported that 'self-concept' had the lowest correlation, followed by 'academic anxiety' and then 'self-efficacy'.[13] A fourth self-belief, self-confidence, is reported to be the best 'non-cognitive' predictor of future academic performance.[13 14]

The measurement of 'confidence' can be separated into two conceptually distinct approaches. First, 'self-confidence' as a trait, is usually captured via responses to self-report questionnaires. Such self-reported 'confidence' may be considered a relatively stable personality trait.[15] Interestingly, higher scores on a questionnaire-based measure of self-confidence among medical school applicants have been shown to be associated with the reduced risk of an individual subsequently reporting experiencing health-related issues at medical school.[16] Second, 'self-confidence' can be conceptualised as an ability to accurately (or otherwise) appraise one's own ability at a certain task.[17] This is also often referred to in the literature as 'online confidence', frequently characterised by how well a test taker is able to judge their own abilities on a written assessment. In this respect, online confidence could be conceptualised as a 'meta-cognitive' attribute (ie, the ability to 'think about thinking'). That is, one which requires a cognitive judgement about one's own cognitive performance.[18] In this sense the term 'non-cognitive', when used in conjunction with this trait, must be used cautiously, as there are suggestions that such self-appraisal has a cognitive component.[14]

The UK Clinical Aptitude Test (UKCAT),[19] subsequently renamed the University Clinical Aptitude Test (UCAT), introduced in 2006, is currently used as a component of selection by the majority of UK medical schools. It presently consists of four cognitive scales ('abstract reasoning', 'decision making', 'quantitative reasoning' and 'verbal reasoning') and a situational judgement test (SJT). The UKCAT Consortium replaced the 'decision analysis' subtest with the 'decision making subtest' in 2017, having piloted items from the latter in 2016. This change was made for several reasons. Given the overall ability level of the candidates it was desirable to have a subtest that discriminated more precisely at the upper end of performance (ie, a test that was experienced as fairly challenging). Moreover, the design of the decision analysis subtest constrained trialling of new items. This led UKCAT (now UCAT) to have had concerns regarding overexposure of test content to potential candidates. It was also hoped that the decision making subtest would assess a relatively broader range of traits relating to decision making, as defined in the Selecting for Excellence Report.[20]

The 'decision analysis' component of the UKCAT was based on a decoding task. Candidates were presented with a scenario and a set of codes (eg, A='Never', B='Bad', C='Lawyer', D='Faithful', E='Employer', F='Good', G='Friends', and so on). The test items consisted of a series of statements (eg, 'Bad lawyers are never loyal to their firm'). The test taker was then presented with a multiple choice of codes (eg, 1. 'ABCDE', 2. 'ABDEF', 3. 'ACDEF', 4. 'ABCEG'). The code that best reflected the meaning of the statement had to be selected from this list by the test takers (in this example 'ABCDE'). For other items the decoding task was reversed and candidates had to select the most appropriate statement from a presented selection, having been given the code. Previous independent analysis demonstrated that the decision analysis had an acceptable level of reliability with Cronbach's alpha (binary version) and McDonald's omega values of 0.87.[21]

Between 2013 and 2016 the UKCAT was used to pilot an online 'confidence rating' within the 'decision analysis' section of the test. After each of the items presented in this section, candidates had to indicate their confidence that the answer they had provided was correct, using a scale from 1 to 5. It was made clear to the candidates at the time that their responses were not going to be used in the selection process. An example item and corresponding confidence rating is shown in figure 1. As can be seen, no guide or anchor points were provided to the candidates for the confidence rating.

Previous work focussed on measuring online confidence has tended to take a relatively simplistic approach to calculating an estimate of this ability. This usually involves creating a 'confidence bias' score, which is defined as the average difference between the confidence self-ratings and the performance on the items.[22] For example, on a 5-point confidence rating scale a test taker may choose the fourth point (eg, 'fairly confident I got the question right') which equates to a score of 0.75 on a scale of 0 to 1 (eg, 'not at all confident' to 'almost sure'). Candidates, as is usual, would score 1 point for a correct answer and 0 for an incorrect response. In order to calculate confidence bias the confidence self-rating would be subtracted from this score. Thus, in this example, a candidate would

be allocated a score of 0.25: that is a score of 1 for a correct answer, minus a rating of 0.75 for a self-rating of 4/5 on the 5-point confidence scale. Had the candidate actually answered the item incorrectly the score would have been −0.75 (ie, 0 for a wrong answer minus 0.75 for the confidence rating). The overall 'confidence bias' for each candidate is then calculated as the mean confidence bias score for all the items the test taker responded to. This means that the theoretical range of confidence bias scores is from −1 to 1, where negative values suggest a tendency for overconfidence and positive values suggest underconfidence in one's own performance on the test.

'Confidence bias' calculated in this manner has previously been reported to be inversely correlated with cognitive ability. Specifically, individuals who tend to report higher levels of confidence on a task have, on average, poorer performance.[22] However, such an association may be vulnerable to potential confounding, given how this metric is derived. That is, an inverse relationship between 'confidence bias' and cognitive performance scores could, at least partly, be an artefact of the score achieved in the latter. Consequently, the confidence bias scores will be heavily dependent on performance on the cognitive task itself. Specifically, in this case, high bias scores are much more likely to be observed in those with relatively poor performance on the decision analysis subtest; thus, a higher frequency of incorrect responses will almost inevitably lead to more opportunities for 'overconfidence' to be observed. Second, it is well recognised that individuals vary in their underlying tendency to respond to questionnaires by choosing central or extreme points on such scales. This is referred to as 'response style' and can be crudely categorised into a 'central tendency' or 'extreme response style', depending on the preference for mid-points or outer regions of scales respectively. A number of approaches have been suggested to adjust for response style.[23] Third, online confidence, as an ability, could be confounded with the self-confidence as a trait if the mean (baseline) levels of confidence are not taken into account. Confidence, in this context could be considered an aspect of metacognition, specifically an ability to evaluate and reflect on one's performance on a task. This contrasts with a trait model of confidence, which would postulate that individuals have a generalised and consistent tendency to take either a positive or negative view of their own abilities. Thus any method to evaluate online confidence must attempt to differentiate between the 'trait' and 'ability' aspects of the construct.

Classical test theory relies on the raw summed scores to estimate the ability of a test taker on an assessment measuring that trait. Thus, the sum of correct responses is assumed to be a 'sufficient statistic' by which to differentiate individuals in that respect. This assumes that all the test items are equally good at discriminating between candidates of differing abilities. In contrast, Item Response Theory (IRT) provides a more sophisticated approach to modelling the ability of a test taker on an assessment and is able to accommodate items of differing

discrimination.[24] This permits a more nuanced metric of online confidence to be derived by evaluating the correlation between the prediction, derived from an IRT model, that a candidate would answer a particular item correctly, and their own judgement regarding whether they felt they had answered the question correctly. Thus, it would be anticipated that candidates with a well-developed ability to appraise their own test performance would have relatively high, positive, correlations between these two values. Conversely individuals with only a limited ability to judge their own ability would show little or no correlation. At extremes, theoretically, very overconfident candidates may show negative correlations between these two elements. We refer to this novel approach of measuring online confidence as estimating 'confidence judgement' in order to differentiate it from the conventional measure of 'confidence bias'. Thus, we define confidence judgement as the accuracy of one's own judgement in relation to one's ability at an online or written test. It can thus be conceptualised as an aspect of metacognition.

The presence of the UKCAT data relating to online confidence allowed us to conduct a study with the primary aim of evaluating the potential for the measurement of this construct to enhance medical selection. This included appraising the potential impact on the demographics of the UK medical student population if implemented. A secondary aim was to compare the properties of our novel metric of online confidence compared with the conventional approach. The study objectives were thus:

1. To evaluate the relationship between two measures of online confidence and the sociodemographic and educational characteristics of medical school applicants.
2. To model the relationship between online confidence and the likelihood of success at medical school application.
3. To model the relationship between online confidence and, for those successful applicants entering UKCAT consortium medical schools, academic performance in the first 2 years of undergraduate study.

The exploration of these data also allowed us to consider the practicalities of implementing such a testing approach to medical selection.

## METHODS
### Data availability and preparation
Applicants to UKCAT consortium medical schools must sit the UKCAT test during the calendar year of their application. The test may only be taken once per application cycle. Data were provided by UKCAT for 65691 candidates who sat the test from 2013 to 2016. Raw data were placed by UKCAT in the Health Informatics Centre — a 'Safe Haven' hosted by the University of Dundee.[25] The Safe Haven is a set of secure servers that allows management and analysis but where, for security, no individual data may be extracted, but only reports on aggregated data or the results of analyses may be outputted. Thus

**Table 1** Numbers of candidates with data available for the analyses.

| Variable | Number of candidates with valid data |
|---|---|
| UKCAT scores | 65691 |
| 'Valid' confidence ratings | 56785 |
| Gender | 56785 |
| Socioeconomic background | 49943 |
| School type attended | 10053 |
| English language status | 55212 |
| Secondary (high) school qualifications | 11384 |
| University application information | 18985 |
| Academic outcome year 1 medical school | 1252 |
| Academic outcome year 2 medical school | 854 |

UKCAT, UK Clinical Aptitude Test.

the data were subsequently cleaned, managed, linked and analysed by the research team inside the Safe Haven. Table 1 and figure 2 show the data that were available for this study.

### Predictor variables

Approximately 10% of the candidates who completed the confidence ratings did not show any variation in their responses (ie, they responded by choosing all ones, or all threes, and so on). These responses were deemed 'invalid' responses and were not further analysed. This left 56785 candidates with 'valid' confidence ratings.

'Confidence bias' for each candidate was calculated as follows. For each item, the confidence rating (on a 1

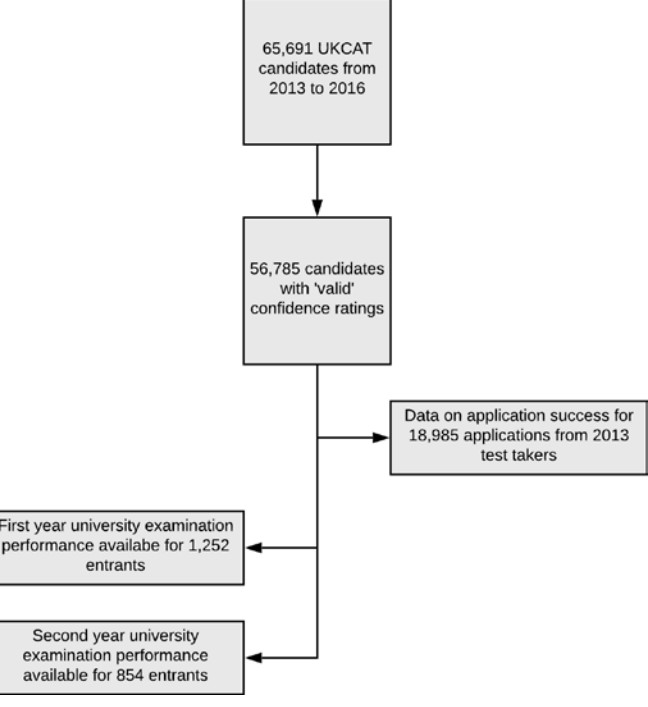

**Figure 2** Data flow chart for the study. UKCAT, UK Clinical AptitudeTest.

to 5 scale) was converted to a confidence score on the unit scale (ie, 0 to 1). This confidence score was then subtracted from the item score (0 for an incorrect answer, 1 for a correct answer). The results were then averaged within candidate to produce the 'confidence bias' score for each applicant, with higher scores representing relative underconfidence.

The novel 'confidence judgement' estimate for each candidate was calculated as follows. For each candidate, a 'two parameter logistic regression IRT' model was used to predict the probability (or to be precise, the log of the odds — 'log odds') of the candidate getting each item correct.[24] This probability is determined by three factors. First, the candidate's ability, which is estimated using their performance across all items of the test, compared with the other individuals in the sample. Second, the item's relative difficulty, as estimated using the performance of all candidates on that specific item. Finally that particular question's 'discrimination' (ie, precision in differentiating between candidates at the ability level suggested by the item's difficulty). Once the probability (ie, log odds) of each candidate getting each item correct had been estimated, the 'confidence judgement' for each candidate was calculated by computing the correlation between that candidate's log odds of getting each item correct and their own confidence rating. Note, in this context the use of the log odds was preferable to the exponentiated OR or probability, as the former behave in a more linear fashion than the latter two values. In order to differentiate the 'ability' aspects of online confidence from the 'trait' element (see earlier) the confidence self-ratings for each item were first adjusted by rescaling them in terms of their mean values ('elevation') and variance ('scatter') for each individual. This method has been previously shown to be useful for such purposes.[26] In practice, such adjustment was done via the use of 'within person z-scores', which standardised the confidence self-ratings within each test taker by subtracting the mean rating and dividing by the SD. Thus each score represented a correlation coefficient for that individual. In this sense '1' would represent a perfect positive correlation between the respondent's estimation of their own ability at the test (ideal judgement), '0' as no correlation (ie, no relationship) and '−1' as a perfect inverse correlation (ie, the respondent was most confident about the items they were least likely to answer correctly).

### Background variables

We also had access to information relating to candidate sociodemographic characteristics (see table 2). In line with previous research[3] we dichotomised gender, socioeconomic status (NS-SEC rating[27] 1 to 3 vs 4 or 5), secondary school type attended (non-selective school, selective school (including state grammar schools)) and language status (native English speaker, English as a second language). As in previous studies relating to the UKCAT, a continuous metric of academic achievement in secondary (high) school was derived.[5] This metric

**Table 2** Confidence bias (CB) and confidence judgement (CJ) scores according to self-reported ethnicity. CB scores less than 0 represent overconfidence, with larger negative values indicating more overconfidence. CJ scores represent the ability for candidates to appraise their own performance, with a CJ of 1 indicating perfect judgement.

| Self-reported ethnicity | Mean CB score (SD) | Mean CJ score (SD) |
|---|---|---|
| White (n=23 679) | −0.01 (0.19) | 0.32 (0.24) |
| Asian (n=13 421) | −0.04 (0.22) | 0.28 (0.25) |
| Black (n=3260) | −0.08 (0.22) | 0.28 (0.24) |
| Mixed (n=1741) | −0.02 (0.21) | 0.30 (0.24) |
| Other (n=1098) | −0.05 (0.22) | 0.29 (0.24) |

consisted of A-level performance and Irish and Scottish qualifications. In the UK, all secondary school grades are associated with a corresponding Universities and Colleges Admissions Service (UCAS) tariff. The metric of prior academic achievement was derived as the percentage of the maximum available UCAS tariff each candidate achieved. Only the three highest grades achieved were included. Any resits were excluded, as were qualifications in 'General Studies'. Standardised z-scores (ie, mean of 0, SD of 1) within nationality were then calculated.

UKCAT scores for each candidate were available. This included overall performance ('UKCAT total score') and performance on each of the four subscales of the test. These five scores were standardised as z-scores according to the UKCAT year of sitting in order to allow cross-comparison across application year. Previously it has been shown that the UKCAT scores can be conceptualised as being divided into those related to verbal and non-verbal reasoning.[21] The UKCAT total scores were made up of three non-verbal subtest scores (quantitative reasoning, abstract reasoning and decision analysis) and one verbal subtest (verbal reasoning). Therefore in order to obtain an overall estimate of cognitive ability a 'balanced' summary form of the test performance was derived by averaging the three standardised scores from the non-verbal subtests, adding the standardised verbal reasoning score and dividing this total by two.

### Outcome variables

Data on whether an application made by a candidate resulted in an offer being made (or not) to study medicine at the university applied to were available for 2013 test takers only (18 985 applications). In this respect information was only available for medical schools in the UKCAT consortium, though this accounted for almost all the UK universities offering medicine courses at that time — that is roughly 30 of 35 medical schools (the numbers varied slightly during the study period).

In order to enter medical school, candidates must both receive an offer from a university and subsequently meet the conditions of that offer (eg, achieving a certain level of secondary (high) school achievement). Of those who

entered medical school, we also had access to performance in knowledge-based and skill-based end of year exams in the first year of university. These data were available for 1252 and 854 candidates, respectively. In order to allow cross-comparison across years and medical schools, these outcomes were standardised as z-scores according to the year of examination and institution of the candidate using the approach employed by previous studies using these outcomes.[3 9 28]

### Data analysis

The data were placed in a Safe Haven, cleaned, managed and linked. All analyses were performed in the Safe Haven using Stata V.14.[29]

Univariable analyses between the confidence ratings and sociodemographic data were analysed via correlations, regressions and Kruskal-Wallis tests. Informed by these analyses, multivariable forwards stepwise regression models were built in order to predict the odds of success at application and the odds of passing the first year of university at the first attempt. As each candidate could apply to multiple medical schools, when looking at the relationship between the confidence scores and the odds of an offer, multilevel logistic regression was used, with offers conceptualised as nested within candidates. As missing data, in terms of item responses and sociodemographic variables, were relatively uncommon, we used listwise deletion to account for missingness. It should also be highlighted that as the predictor variables (UKCAT subtest scores etc) were standardised according to the applicants, then the resulting regression or correlation coefficients did not require the usual correction approaches for the restriction of range due to the selection procedure (ie, only applicants that were successful had academic outcomes observed).[30] Obtaining academic outcomes relied on the medical schools in the UKCAT consortium returning them annually. Thus, the missing data related to the academic outcomes were assumed to be either 'missing completely at random' (ie, purely due to chance) or 'missing at random' (related to observed variables) and thus unlikely to threaten the validity of the results. Previously, sensitivity analyses provided evidence that these academic outcomes, particularly in the first 2 years of undergraduate medical education, were likely to be largely missing in this way.[3]

In order to evaluate the extent that extreme response style may influence or confound confidence bias scores, we estimated this trait separately. This was done by a previously outlined method whereby 'shadow indicators' were created (0 or 1) depending on whether an 'extreme' response was selected (in this case 1 or 5).[23] This trait was separately modelled for each candidate using a two parameter logistic IRT model with the 'shadow indicators' as binary indicators.

### Patient and public involvement

There was no patient or public involvement in this study.

## RESULTS

As might be expected, there was at least modest correlation (r=0.24) between 'confidence bias' and 'confidence judgement'. The mean 'confidence bias' score was −0.04 (SD 0.21) and the mean 'confidence judgement' value was 0.30 (SD 0.24). Reliability indices, in terms of Cronbach's alpha, for the 'confidence bias' values related to the constituent items for the five forms of the test ranged from 0.83 to 0.85. An equivalent reliability metric could not be calculated for confidence judgement as it is actually a 'within person' (rather than 'within test') correlation coefficient.

### Relationship between online confidence and sociodemographic data

Kruskal-Wallis testing showed that lower 'confidence bias' scores (ie, overconfidence) were significantly (p<0.001) associated with male gender, English as a second language (EASL), self-declared non-White ethnicity, non-professional socioeconomic background and attendance at a non-selective secondary school. When controlling for score achieved on the 'decision analysis' subtest, this relationship remained significant (p<0.01) only for male gender, EASL and non-selective schooling. Similarly, poorer 'confidence judgement' (ie, confidence ratings were less highly correlated with the actual probability of answering a question correctly) was observed for males, those with EASL, candidates from a non-professional socioeconomic background and those reporting non-White ethnicity (p<0.0001 in all cases). These associations remained significant at the p<0.001 level after adjusting for performance on the 'decision analysis' subtest with the exception of non-professional background (p=0.4). 'Confidence judgement' was not associated significantly with secondary school-type attended (p=0.2).

Table 2 depicts the relationships between the two online confidence metrics and self-reported ethnicity. As can be seen, those self-reporting as 'White' display the lowest levels of overconfidence (as measured by 'confidence bias') and also display the highest levels of accuracy in their own ability, as indexed by 'confidence judgement'. The differences between self-reported ethnic groups appears somewhat more marked for the 'confidence bias' scores, compared with the 'confidence judgement' estimates. However, in the former case these differences become statistically non-significant once performance on the decision analysis subtest are controlled for. For example, the most marked difference in both online confidence metrics can be seen between those reporting themselves as of 'white' ethnicity and those self-identifying as 'black'. Self-reported ethnicity ('white' vs 'black') can be predicted from confidence bias score (OR 0.18, 0.15 to 0.22, p<0.001). However, once the influence of performance on the decision analysis subtest is controlled for, this relationship with 'confidence bias' score disappears (OR 1.04, 0.84 to 1.29, p=0.7). Thus, the inter-ethnic group differences for 'confidence bias' appear to be an artefact of performance on the related cognitive test (ie,

**Table 3** Correlation between educational performance (standardised A-level tariff and UK Clinical Aptitude Test (UKCAT) performance) and confidence ratings (confidence bias (CB) and confidence judgement (CJ)).

| Variable | Correlation with CB (n=56 785) | Correlation with CJ (n=56 479) |
|---|---|---|
| Standardised secondary school qualification tariff | 0.16 | 0.06 |
| Standardised 'decision analysis' score | 0.51 | 0.16 |
| Standardised 'abstract reasoning' score | 0.23 | 0.07 |
| Standardised 'quantitative reasoning' score | 0.23 | 0.09 |
| Standardised 'verbal reasoning' score | 0.21 | 0.09 |
| Standardised total UKCAT score | 0.39 | 0.14 |
| 'Balanced' UKCAT score | 0.34 | 0.12 |
| Extreme response style | −0.46 | −0.01 |

decision analysis). A similar magnitude of univariable relationship is observed between these two self-reported ethnic groups and 'confidence judgement' (OR 0.49, 0.42 to 0.57, p<0.001). However, the relationship remains statistically significant even after the influence of performance on decision analysis is accounted for (OR 0.74, 0.64 to 0.87, p<0.001).

### Relationship between online confidence and prior academic performance

As can be seen from table 3, both 'confidence bias' and 'confidence judgement' correlate positively with standardised secondary school performance and standardised UKCAT performance. The correlation between confidence bias and the standardised decision analysis scores was moderate (r=0.51). This highlights the dependency of this traditional metric of 'online confidence' on task performance (see also earlier, in Methods section). In contrast a more modest correlation between this index of overall performance on this cognitive subtest and 'confidence judgement' was observed (r=0.16), suggesting it is less dependent on performance at the related cognitive task. Similarly, unlike confidence bias, there was no relationship observed between extreme response style and confidence judgement.

### Online confidence and success at application to medical school

Mann-Whitney U testing showed small yet statistically significant differences between the 2013 cohort (the only cohort for which we had data on application success) and later cohorts on online confidence and the 'decision analysis' scale.

Univariable multilevel logistic regression showed that those with higher 'confidence bias' scores (ie, those who are relatively underconfident) were much more likely to receive an offer to study medicine (OR 11.97, 9.55 to 15.01, p<0.001). This can be interpreted as follows: for

**Table 4** Results from the multivariable multilevel models predicting an offer from 'confidence bias' scores in candidates (n=7870), controlling for the influence of other statistically significant predictor variables.

| Predictor variable | OR | 95% CI for OR | P value |
|---|---|---|---|
| Confidence bias | 1.48 | 1.15 to 1.91 | 0.002 |
| Male gender | 0.69 | 0.64 to 0.76 | <0.001 |
| Standardised 'decision analysis' score | 1.49 | 1.39 to 1.60 | <0.001 |
| Standardised 'verbal reasoning' score | 1.51 | 1.43 to 1.59 | <0.001 |
| Standardised 'quantitative reasoning' score | 1.40 | 1.32 to 1.49 | <0.001 |
| Standardised 'abstract reasoning' score | 1.44 | 1.37 to 1.51 | <0.001 |
| Standardised secondary school qualification tariff | 1.48 | 1.39 to 1.58 | <0.001 |
| 'Non-white' ethnicity | 0.89 | 0.81 to 0.97 | 0.012 |
| Non-selective secondary (high) school attended | 0.92 | 0.84 to 1.00 | 0.046 |

every SD above the mean an applicant's 'confidence bias' score was, their odds of receiving an offer to study medicine increased by a factor of 12. After controlling for background variables, including cognitive ability, in a multivariable stepwise regression, underconfidence remained a statistically significant independent predictor of receiving an offer from a medical school (OR 1.48, 1.15 to 1.91, p=0.002). The full results of this model are shown in table 4.

'Confidence judgement' was also a univariable predictor of application success (OR 1.22, 1.01 to 1.47, p=0.04), although of only borderline statistical significance. However, in contrast to 'confidence bias', once the potential influence of cognitive ability and background variables were controlled for 'confidence judgement' was not an independent, statistically significant, predictor of receiving an offer (OR 0.98, 0.80 to 1.20, p=0.8).

### Online confidence and academic performance during medical school

Mann-Whitney U testing showed that those individuals with data on academic performance in medical school had statistically significantly higher confidence bias, confidence judgement and decision analysis scores than the data set as a whole.

The only statistically significant association between 'confidence bias' and undergraduate performance was that observed for the odds of passing the end of year 1 at the first attempt (OR 4.37, 1.54 to 12.42, p=0.006). That is, those who reported relative underconfidence (as indexed by averaging one point above the mean on the 'confidence bias' score) had over four times the odds of passing the year at first attempt, compared with those with average scores. This effect remained statistically significant when controlling for performance on

the 'decision analysis' subtest of the UKCAT (OR 3.24, 1.08 to 9.72, p=0.04). As the outcome was categorical, no easily interpretable R statistic (representing the proportion of variance explained by the predictor variables) was available, in contrast to continuous variables modelled using linear regression. However, an analogous statistic for logistic regression exists in 'McFadden's pseudo-$R^2$'.[31] This reflects the amount of variance explained in the hypothetical latent variable, postulated to be underlying the observed responses. In the case of predicting the odds of passing year 1 at first attempt both performance on the decision analysis subtest and the confidence bias scores had relatively low pseudo-$R^2$ values, though the addition of the latter into the logistic regression equation doubled its magnitude. Specifically, the pseudo-$R^2$ value for predicting the odds of passing year 1 at first attempt from the standardised decision analysis score was 0.0057. This increased to 0.011 when the confidence bias scores were entered. Thus, at least for this specific academic outcome the tradition measure of online confidence appeared to show some incremental predictive validity over the linked cognitive task.

Once the potential influence of other background variables were controlled for, 'confidence bias' scores were no longer significantly predictive of the odds of passing year 1 of medical school at the first attempt (OR 1.96, 0.43 to 9.03, p=0.4). We observed no significant relationships between 'confidence bias' score and the odds of passing year 2 at first attempt, nor with performance on knowledge or skills-based assessments in either of the first 2 years of medical school.

The relationship between 'confidence judgement' and subsequent academic performance was even weaker. We observed no statistically significant associations, or even modest trends, between confidence judgement scores and any of the academic outcomes we had access to.

### DISCUSSION

In these analyses we examined the relationship between 'online confidence', sociodemographic data, the probability of an offer of a place to study medicine and subsequent academic performance in medical school. We were able to use an existing metric of online confidence, as utilised in the original pilot study and provisional descriptive analysis by Pearson VUE.[32] In addition we were able to explore the use of an experimental method to evaluate 'confidence judgement', using a relatively sophisticated approach to modelling self-appraisal of ability at a cognitive test.

In line with previous research we found that overconfidence was inversely related to cognitive performance.[14] In this study we observed that underconfidence, as measured by the 'confidence bias' score, was associated with an increased odds of receiving an offer to study medicine. Moreover this effect remained significant once the impact of their background demographic and academic variables were taken into consideration. Additionally,

'confidence bias' was modestly linked to subsequent academic performance, in the sense that those that were relatively underconfident had higher odds of passing the first year of undergraduate medicine at first sitting. This association was relatively independent of cognitive ability, though not of the influence of sociodemographic background variables.

The novel measure of 'confidence judgement' showed somewhat different properties to that of 'confidence bias', despite the scores derived from the two approaches correlating moderately. 'Confidence judgement' showed a much weaker relationship with cognitive ability, which is undoubtedly advantageous when trying to delineate between cognitive ability and this potential aspect of metacognition. There was also a somewhat different relationship with self-reported ethnicity. Specifically, intergroup ethnic differences were not quite as marked for 'confidence judgement', compared with 'confidence bias'. However, in the case of the latter, the observed relationship appeared to be totally accounted for by a candidate's performance on the decision analysis subtest. This was not so with the 'confidence judgement' measure, which may be picking up genuine differences between self-reported ethnic groups in terms of metacognitive style.

Importantly, in contrast to 'confidence bias', there were no observed relationships between 'confidence judgement' and undergraduate academic performance. However 'confidence judgement' was significantly related to the odds of receiving an offer from medical school, though this effect was not independent of other cognitive, educational and background factors.

### Possible interpretations

The association between underconfidence and an increased chance of receiving an offer for medical school was independent of other educational, cognitive and sociodemographic factors. It could be that candidates who appear overconfident are less likely to receive offers since the vast majority of UK medical schools still use face-to-face interviews and/or group exercises as part of the selection process.

'Confidence judgement' had lower observed correlation with cognitive tasks than 'confidence bias'. This suggests that 'confidence judgement' (ie, a candidate's judgement of their own ability) is less of a proxy for overall ability at a cognitive task than 'confidence bias' (whether a candidate is on average over or under confident). Furthermore, 'confidence bias' may be measuring other irrelevant constructs, such as 'extreme response style' (see table 3). It may also be that the 'confidence judgement' estimates are less prone to confounding with general intellectual ability. Indeed, it could be conceptualised as a more nuanced metacognitive skill focussed on being able to appraise the relative difficulty of test items in relation to one's own ability in the domain being tested.

It could by hypothesised that the lower observed correlations for 'confidence judgement' are that it is less 'reliable', in some sense. As 'confidence judgement'

is itself an individual correlation coefficient, traditional measures of reliability are not applicable. However, it is possible that 'confidence judgement' captures relatively less information on each candidate than 'confidence bias' does. However, we note that we did observe significant associations between 'confidence judgement' and the odds of receiving an offer to study medicine, as well as differences in the measure across ethnicities. This suggests the measure is able to discriminate between individuals. Nevertheless, we cannot rule out that the lower correlations observed in some cases are due to 'lower reliability'.

It could be hypothesised that future doctors should ideally show neither overconfidence nor underconfidence. However, we did not observe any non-linear effects for 'confidence bias' in our analyses. This would suggest that no such 'sweet spot' exists, at least in relation to the outcome measures examined in this study. Indeed, in this context, the facet of overconfidence or underconfidence may be better conceptualised as a component of interpersonal competency. That is, individuals who are more narcissistic, and hence may be 'overconfident' are likely to encounter more interpersonal difficulties in the workplace.[33] However, recent findings have (depressingly) suggested that narcissism, as a trait, may be associated with greater academic success than an individual's cognitive ability might otherwise suggest.[34] On the other hand, one study reported that higher self-rated confidence in medical undergraduates was associated with poorer academic performance.[9]

We also noted that all standardised subtest scores on the UKCAT were independent and statistically significant predictors of the odds of an offer of a place to study at medical school, with very similar effects sizes (table 4). Presumably this observation reflects the common use of the UKCAT summed total score within the admissions process, which would, in effect, give each component an equal weighting.

Formal testing showed statistically significant differences between the 2013 cohort and later cohorts. However, the actual effect sizes were trivially small and may have been caused by differences in the demographics.

### Implications for policy

In theory, the ability to evaluate underconfidence or overconfidence, as distinct from cognitive ability, in medical applicants should add value within the selection process. In particular the construct could predict more distal, non-academic, aspects of performance. There would, however, appear to be a number of practical challenges with implementing either 'confidence bias' or 'confidence judgement' assessments in high stakes selection. First, it was noted in these data that a proportion of candidates did not vary in their responses in relation to their perceived confidence (ie, there was zero variance). Thus such candidates provided no information about how they perceived their confidence related to their actual ability. It is difficult to see how such invalid response patterns

could be discouraged or mitigated in practice. Indeed, if candidates believe that underconfidence was more desirable than overconfidence, they may deliberately rate their confidence rather lower than they might otherwise. It is difficult to safeguard against such social desirability bias. Moreover, overall, there seemed to be a stronger relationship between online confidence and success at application than with subsequent academic performance, though this could have been partly an artefact of 'restriction of range' as a result of the selection process.[35] This would have been mainly, statistically, addressed via standardising the predictor variables according to the applicant, rather than entrant pool.[30] Thus, it would seem unwise to implement a selection measure where success at application was not reasonably mirrored by performance on subsequent work-related metrics. There would also be challenges, related to the wider context to how the resulting scores might be interpreted and used by selectors. For example, would the ratings be treated as a criterion or norm referenced measure? How might the demographic associations with online confidence, highlighted in the results of the present study, be handled, to prevent certain population groups being disadvantaged?

When considering the introduction of new metrics into the selection process, it is important that they add value above and beyond those metrics already in use. A particular problem with using the conventional estimate of 'confidence bias' is that it correlates at least moderately with actual cognitive ability in the domain tested (in this case r=0.51). Therefore, there are some doubts about the incremental value that 'confidence bias' ratings would add to the selection process, at least in terms of academic performance. Nevertheless, the estimation of online confidence may still prove useful. It could still be used in selection if the practical challenges of measuring online confidence in a high stakes setting could be overcome. There may also be potential to use online confidence as a future research tool, exploring the link between self-appraisal and actual clinical practice in medical staff. With the increased focus on assessing the interpersonal competence of medical applicants it may also be worth revisiting some of the earlier work around 'emotional intelligence' as an applied ability.[36] That is, being able to accurately identify emotional states in oneself and others, and also being able to respond to them effectively. Such traits can be evaluated, to some extent, via situational judgement tests, though more resource intensive approaches may be required to increase the precision by which such 'non-cognitive' attributes are measured. Thus, it may be more desirable, from a personnel selection perspective, to focus on the personal qualities associated with trait confidence and interpersonal effectiveness, in contrast to online confidence, as an aspect of metacognition.

## Potential strengths and limitations

There were three key limitations with this study. First, self-report confidence measures have previously been piloted as part of the UKCAT.[9] However, we could not link the data relating to online confidence presented in this paper to the self-reported confidence, as there was a negligible overlap between candidates who had undertaken both the measures. Thus, we could not compare confidence as a trait and as a metacognitive ability.

Second, while confidence has previously been explored as a predictor of academic performance, one could argue that it would be of most interest to link such measures to interpersonal functioning, such as fitness to practise issues. However at present, such outcomes are not freely available. In the future it may be possible to link measures of confidence to these outcomes. However fitness to practise issues are a relatively infrequent occurrence and therefore study power would remain challenging. However, it may be feasible to link metrics of confidence with other outcomes related to interpersonal functioning, such as 'high fidelity' simulations featuring role played patients. It is possible that these first two limitations could be addressed if the data available here were included in the UK Medical Education Database,[37] the UK's national data repository for medical trainees.

Third, these tests were piloted in low-stakes conditions. It is thus unclear the extent to which they may generalise to high-stakes settings. Indeed, although it is indisputably desirable to select candidates on personal qualities beyond cognitive and academic ability, there are substantial challenges with evaluating such traits in a high-stakes setting.[10] The most obvious threat to the validity of the scores from such assessments is posed by faking-effects or social desirability bias. However, there are also some deeper philosophical issues, which surfaced in previous debates relating to the overall concept and measurement of 'emotional intelligence', of which the ability to appraise one's own ability could be considered a putative facet. Indeed, there were two main criticisms of the concept of 'emotional intelligence' as an ability. First, tests of such personal qualities tried to capture 'typical performance', that is an individual's general behavioural tendency. This is in contrast to tests of 'maximum performance', such as those related to academic or cognitive ability, where a test taker would attempt to achieve the highest score possible.[38 39] Another related criticism of the idea that such personal qualities could be conceptualised as abilities was the problem of deriving a scoring rubric. That is, tests evaluating such traits generally cannot be considered to have 'right' or 'wrong' answers, in the same way that cognitive assessments do. This leads to difficulties in deriving valid and fair scoring systems, with all options, including expert and normative-based keys having flaws.[40]

Other limitations to this study are worth noting. These include the fact that only applications to UKCAT consortium universities could be observed, though these represented most medical schools at the time of the study. The number of UK medical schools operating in the UK varied slightly during the study period (for example, Aston and the University of Central Lancashire both launched in 2015). However, the vast majority of medical schools were part of the UKCAT consortium at that time, with roughly

30 of the approximately 35 British medical schools having membership. However, the few universities using the Biomedical Admissions Test (BMAT), the main alternative selection assessment, included Oxford and Cambridge. Therefore, it is not possible to rule out that those who apply solely to non-UKCAT consortium medical schools are more or less confident than those applicants present in this study. Nevertheless, our experience of working with medical selection assessments is that almost all candidates who sit the BMAT also sit the UKCAT, though the converse is not true. This is presumably the case because medical school applicants would generally not want their options at application limited to the small number of universities that use the BMAT as their selection assessment. As with previous studies relating to the UKCAT, a reliance on local, medical school assessments, where the definition of 'knowledge' and 'skills' based assessments are not operationalised and defined, limits the inferences we can draw from such results.[3 28]

## CONCLUSIONS

We observed relatively few associations between online confidence in medical applicants and the outcomes of interest, though underconfidence seemed a relatively robust predictor of success at application. However, implementing such measures in high-stakes selection situations may not be feasible or desirable at present. There are issues in relation to both the practicality of such an approach as well as questions related to the added value that such a metric may add to the current battery of tests already employed by many medical schools in Western countries. Thus, in practice it may be more fruitful to concentrate on the attempted measurement of other aspects of personal qualities that may be deemed important to future clinical performance, in particular those that are less strongly related to cognitive ability.

**Acknowledgements** We thank Rachel Greatrix from UKCAT for her assistance with accessing the data used in the study. We are also grateful to Brad Wu from Pearson VUE for supervising the original pilot and his helpful comments on an earlier version of this manuscript.

**Contributors** PAT lead on study conception, data analysis and drafting the manuscript. LWP contributed to data analysis, interpretation and drafting the manuscript. Both authors approve the manuscript and agree with its submission to BMJ open. Both authors agree to be accountable for all aspects of the work.

**Funding** PAT receives funding for his academic time via an NIHR Career Development Fellowship. This paper represents independent research part-funded by the National Institute for Health Research (NIHR). The views expressed are those of the authors and not necessarily those of the NHS, the NIHR or the Department of Health. The UKCAT Board provided funding for a portion of LWP's time on this project.

**Competing interests** PAT has previously received research funding from the ESRC, the EPSRC, the Department of Health for England, the UKCAT board and the GMC. In addition, PAT has previously performed consultancy work on behalf of his employing University for the UKCAT Board and Work Psychology Group, and has received travel and subsistence expenses for attendance at the UKCAT Research Group. LWP has received travel and subsistence expenses for attendance at the UKCAT consortium meeting. The UKCAT Board partly funded this research but did not take an active role in determining the study design or reporting the results.

**Patient consent for publication** Not required.

**Ethics approval** Ethical approval for this study was not required as it relied on the analysis of de-identified routinely collected data. This was confirmed in writing by Durham University's School for Health Ethics Committee.

**Provenance and peer review** Not commissioned; externally peer reviewed.

**Data availability statement** The data used in this study may be made available from UCAT on an application being approved by the UCAT Board.

**ORCID iDs**
Paul A Tiffin https://orcid.org/0000-0003-1770-5034
Lewis W Paton http://orcid.org/0000-0002-3328-5634

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
