## [Reviewer comments · BMJ Open]

ARTICLE DETAILS

TITLE (PROVISIONAL)	Does 'online confidence' predict application success and later academic performance in medical school? A UK-based national cohort study
AUTHORS	Tiffin, Paul; Paton, Lewis

VERSION 1 – REVIEW

REVIEWER	Rob Meijer Rijksuniversiteit Groningen, the Netherlands
REVIEW RETURNED	04-Oct-2019

GENERAL COMMENTS	Thanks, interesting manuscript, well written, well conducted. I have one major remark that you may reflect upon somewhere in the manuscript and may discuss somewhere in the manuscript. As I understand it correctly self-confidence was measured in low-stakes setting because although it was part of a high-stakes selection procedure, the candidates were told that their scores were not taken into account., right? My general problem with this kind of research is that it is very difficult to generalise these findings to the settings where it is high-stake.. I really would like the authors to reflect on this and then take also some publications into account that discuss this in a more general context. See for example Niessen & Meijer (2017, perspectives in psychological science, on the use of broadened admission criteria in higher education). Another idea is the question how much additional variance would be explained by these noncognitive traits above cognitive traits. My general problem is that because this is not "maximum performance" but typical performance and self report, how fair is it (because of all kinds of biases) to select persons on the basis of these measures. Are we selecting the best manipulator as a potential doctor? I would very much like you to reflect on this, thanks for the nice paper!
--

REVIEWER	Katherine Woolf University College London, Research Department for Medical Education
REVIEW RETURNED	07-Oct-2019

GENERAL COMMENTS	This is an interesting and novel paper that uses item response theory to derive a measure of test confidence, and compares it to a standard measure. I'm grateful for the opportunity to review it. The authors are careful to point out the limitations of using measures of confidence in selection, but nonetheless this paper will contribute to the assessment literature. I have a number of suggestions and comments:
---

	1. The abstract should specifically provide results for the CJ measure. 2. I am not clear why so few Year 1 and 2 scores were available. The authors say "missing data were relatively uncommon" (p12) yet there are clearly a lot of structurally missing data due to applicants not getting a place, but I'm guessing there are also other sorts of missing data since there were only Year 1 data on 1252 people. This needs more detailed explanation, and the implications need further discussion. This is important when performance at medical school is a significant outcome measure from a policy perspective. On a much more minor related point, I'm unclear why in the Figure 2 flowchat, the Data on application success" box comes underneath the "First year examination performance" box. 3. UKCAT got rid of the Decision Analysis subtest - why? Was it psychometrically flawed? I would be helpful to give some basic test statistics for each of the subtests, including reliability if available. This seems important since the confidence ratings were for the DA subtest only, and if the test is problematic this could affect the CB and CJ metrics. 4. The correlation between CB and decision analysis scores were high (0.51). Is this because CB is derived from DA scores? If so, this should be made clearer. 5. On p13 I could not understand the third sentence of the Results section. What is "this cognitive subtest"? The DA score? What does "less of a pure proxy" mean? Does it mean that the CJ is measuring something different from the DA, whereas the CB and DA are measuring very similar things (which might not be surprising since the CB is derived from the DA)? This sentence is in fact repeated on p14 (line 26) so perhaps it is just in the wrong place? 6. Can the authors comment on the meaning of the ethnic differences in CB scores but not on CJ scores, in the Discussion? Is this just due to differences in performance on the DA? Or is there more to it? 7. regarding the Extreme Response Style "trait" on p12, I thought the authors removed all the people who gave all "1"s? And perhaps I missed it, but I don't see anything in the Results about the ESR. Why not? 8. Table 4: why did the authors not include other UKCAT subtest scores as predictors? 9. The use of "cognitive ability" is confusing. It would be helpful if the authors could be more specific about when they are talking about DA scores, when they are talking about other measures of cognitive ability. 10. It is really helpful when the authors explain in the text what low or high confidence scores mean. Please do this everywhere! For example p15 line 29. Minor comments: p4 line 12: "accurately estimate self-confidence" should be "accurately estimate ability". p4 line 49: I think "self-efficiency" should be "self efficacy". p5. line 43 to 56 I found it really difficult to understand the description of the DA test. p6. line 26. Remove comma after "Previous work". p7 line 37. What does "online confidence as an ability" mean? Confidence in ability to answer DA questions correctly? I think the authors mean confidence on the DA test vs general confidence. I suspect many readers won't be familiar with the concept of psychological traits, so this needs a bit more explaining.
--	--

	p7 line 41. "lift and scatter" needs more explaining here. Also why lift and not elevate? p8 line 9. "their own judgement (adjusted for 'lift and scatter') " feels like it's missing a few words. Judgement of what? Confidence? p9. line 29. preceding year is confusing. They take the test in the calendar year they apply. p9 line 31 The sentence starting "All data were placed in a safehaven..." is repeated later on. It also needs a bit more detail. Was this done within UKMED? Which safehaven? Who did the linkage and the cleaning? p11. How many medical schools were UKCAT schools at the time? I also think it's possible that BMAT schools are different. This is mentioned in the Discussion but a little more could be made of it. p11. There are two steps between getting an offer and starting at medical school: accepting an offer and meeting that offer. This could be briefly explained before going on to "of those who entered medical school..." (line 55).
--	--

VERSION 1 – AUTHOR RESPONSE

Reviewer(s)' Comments to Author:

Reviewer: 1

Reviewer Name: Rob Meijer

Institution and Country: Rijksuniversiteit Groningen, the Netherlands

Please state any competing interests or state 'None declared': no competing interests

Please leave your comments for the authors below

Thanks, interesting manuscript, well written, well conducted.

Authors' response: Thank you for your positive comments.

Reviewer(s)' Comments to Author: I have one major remark that you may reflect upon somewhere in the manuscript and may discuss somewhere in the manuscript. As I understand it correctly self-confidence was measured in low-stakes setting because although it was part of a high-stakes selection procedure, the candidates were told that their scores were not taken into account., right?

Authors' response: Yes- that is correct.

Reviewer(s)' Comments to Author: My general problem with this kind of research is that it is very difficult to generalise these findings to the settings where it is high-stake.. I really would like the authors to reflect on this and then take also some publications into account that discuss this in a more general context. See for example Niessen & Meijer (2017, perspectives in psychological science, on the use of broadened admission criteria in higher education).

Authors' response: We have expanded the discussion regarding the generalisability of these low-stakes findings to high-stakes selection. Specifically we have included the reference mentioned (Niessen & Meijer, 2017) as well as making some reference to the literature that discussed the 'maximum' vs 'typical' performance issue in relation to the 'emotional intelligence' measurement debate.

Reviewer(s)' Comments to Author: Another idea is the question how much additional variance would be explained by these noncognitive traits above cognitive traits. My general problem is that because this is not "maximum performance" but typical performance and self report, how fair is it (because of all kinds of biases) to select persons on the basis of these measures. Are we selecting the best

manipulator as a potential doctor? I would very much like you to select on this, thanks for the nice paper !

Authors' response: This is a valid point. As the outcomes were categorical then the additional variance explained by the online confidence is not applicable as such. However, in order to address this comment, we now make reference to the logistic regression equivalent- McFadden's pseudo-R2 statistic. Only the 'confidence bias' metric of online confidence showed incremental prediction over performance on the DA subtest. Thus, we mention the McFadden's pseudo-R2 statistics in relation to this finding. Also, when the reviewer asks us to 'to select on this', we presume they meant 'reflect on this', which we have now done.

Reviewer: 2

Reviewer Name: Katherine Woolf

Institution and Country: UCL, UK

Please state any competing interests or state 'None declared': None declared.

Please leave your comments for the authors below

This is an interesting and novel paper that uses item response theory to derive a measure of test confidence, and compares it to a standard measure. I'm grateful for the opportunity to review it.

Authors' response: We thank the reviewer for their positive comments.

Reviewer(s)' Comments to Author: The authors are careful to point out the limitations of using measures of confidence in selection, but nonetheless this paper will contribute to the assessment literature.

I have a number of suggestions and comments:

1. The abstract should specifically provide results for the CJ measure.

Authors' response: Results in relation to 'confidence judgment' have now been added to the abstract.

Reviewer(s)' Comments to Author: 2. I am not clear why so few Year 1 and 2 scores were available. The authors say "missing data were relatively uncommon" (p12) yet there are clearly a lot of structurally missing data due to applicants not getting a place, but I'm guessing there are also other sorts of missing data since there were only Year 1 data on 1252 people. This needs more detailed explanation, and the implications need further discussion. This is important when performance at medical school is a significant outcome measure from a policy perspective.

Authors' response: The original sentence 'missing data were relatively uncommon' was actually referring to missing information relating to the psychometric and sociodemographic data of the participants. This has now been clarified.

We now include an explanation for the missing year 1 and year 1 academic performance scores. That is, more detailed information on academic performance depended on individual medical schools returning these data to the UKCAT Board. Previous sensitivity analysis using multiple imputation suggests that these missing data are either missing at random or missing completely at random (reference now cited). We provide a reference to a previous study where such sensitivity analyses were conducted in order to support this assumption. Thus, in this case, the missing academic outcomes are likely to have impacted on our overall findings and conclusions.

Reviewer(s)' Comments to Author: On a much more minor related point, I'm unclear why in the Figure 2 flowchart, the Data on application success" box comes underneath the "First year examination performance" box.

Authors' response: This has now been corrected. We have also changed the word 'candidate' to 'entrants' for those with university examination outcomes.

Reviewer(s)' Comments to Author:3. UKCAT got rid of the Decision Analysis subtest - why? Was it psychometrically flawed? I would be helpful to give some basic test statistics for each of the subtests, including reliability if available. This seems important since the confidence ratings were for the DA subtest only, and if the test is problematic this could affect the CB and CJ metrics.

Authors' response: This is a valid point. We now included a reference which describes an independent analysis of the UKCAT subtests, in terms of the dimensionality of the test responses and the reliability metrics for each of the tests. We also cite the independently evaluated reliability of the decision analysis subtest.

There were a number of reasons why the decision analysis subtest was replaced by the 'decision-making' test. We have consulted with the UKCAT regarding this. The UKCAT Board is happy for us to put the following sentence in: ""The UKCAT Consortium replaced the 'decision analysis' subtest with the 'decision making subtest' in 2017, having piloted items from the latter in 2016. This change was made for several reasons. Given the overall ability level of the candidates it was desirable to have a subtest that discriminated more precisely at the upper end of performance (i.e. a test that was experienced as fairly challenging). Moreover, the design of the decision analysis subtest constrained trialling of new items. This led UKCAT (now UCAT) to have had concerns regarding overexposure of test content to potential candidates. It was also hoped that the decision making subtest would assess a relatively broader range of traits relating to decision making, as defined in the Selecting for Excellence Report."

Reviewer(s)' Comments to Author:4. The correlation between CB and decision analysis scores were high (0.51). Is this because CB is derived from DA scores? If so, this should be made clearer.

Authors' response: Yes-this is correct. We describe that the confidence bias (CB), the traditionally derived measure of online confidence, is highly dependent on performance on the decision analysis items. This is one of the primary reasons we developed a novel method of estimating online confidence ('confidence judgement'-CJ). We now make this clearer in the methods section. As can be seen from the original Table 3, the correlation between decision analysis performance and confidence judgement is much lower, at 0.16. We now make reference to this point in the results section, as it also highlights the greater robustness of our novel measure of online confidence, compared to the traditional approach for estimating this construct. Note also, for consistency, we have provided the correlation of CJ and DA according to the standardised DA score ($r=0.16$) in the text.

Reviewer(s)' Comments to Author:5. On p13 I could not understand the third sentence of the Results section. What is "this cognitive subtest"? The DA score? What does "less of a pure proxy" mean? Does it mean that the CJ is measuring something different from the DA, whereas the CB and DA are measuring very similar things (which might not be surprising since the CB is derived from the DA)? This sentence is in fact repeated on p14 (line 26) so perhaps it is just in the wrong place?

Authors' response: We agree that this section is not clear. We have now reworded it to make it clear what we are referring to is the dependence of the confidence bias score on performance on the decision analysis score. We've also altered the sentence on page 14, line 26 in the original manuscript, in line with this suggestion, and reordered parts of this section.

Reviewer(s)' Comments to Author:6. Can the authors comment on the meaning of the ethnic differences in CB scores but not on CJ scores, in the Discussion? Is this just due to differences in performance on the DA? Or is there more to it?

Authors' response: Actually both measures of online confidence were sensitive to reported ethnic status to some extent. Looking again at the text, this is not always as clear as it could be. Therefore we have reworded in places to improve the readability. However, the relationship between these two measures and ethnicity was somewhat different. We have included reference to some additional analyses to try and unpack this relationship and explain it to some degree. Specifically, in response to

this point, we show that for CB inter-ethnic differences are accounted for by performance on DA but not so for CJ. We also make reference to this in the discussion section.

Reviewer(s)' Comments to Author:7. regarding the Extreme Response Style "trait" on p12, I thought the authors removed all the people who gave all "1"s? And perhaps I missed it, but I don't see anything in the Results about the ESR. Why not?

Authors' response: These are two separate issues. Participants that gave all '1s' as a responses were assumed not to have engaged as intended with the assessment and thus the responses were invalid. Extreme response style (ERS) is a separate issue. There is an extensive psychometric literature on ERS and we cite at least one paper in this regard. The novel metric of 'confidence judgement' was designed to ameliorate the effects of ERS by adjusting for the 'scatter' of responses within respondents. However, having estimated this behaviour, which can be conceptualised as a trait, characterised by a tendency to use the extremes points on a rating scale, we have now included some reference to it in the results and discussions as appropriate. In this regard we have added the correlation of ERS with the estimates of online confidence in table 3. This shows that ERS (as an estimated trait), as expected, correlates much more highly with confidence bias ($r=-0.46$) than with confidence judgment ($r=-0.01$). Whilst response styles have previously been shown to vary to some extent between ethnic groups, for this sample we only noted a borderline statistically significant difference between those identifying as 'mixed' ethnicity and those categorised as 'other'. Therefore we do not go further into detail about this issue in the paper.

Reviewer(s)' Comments to Author:8. Table 4: why did the authors not include other UKCAT subtest scores as predictors?

Authors' response: The scores between the differing subtests of the UKCAT correlate relatively highly, especially those that measure non-verbal reasoning. Therefore, we did not initially include them in the final multivariable model for this reason. However, we have now included them, and they appear to be each independent and statistically significant predictors of the odds of an offer. The revised results have been put in an amended table 4. We also now make reference to this in the discussion section.

Reviewer(s)' Comments to Author:9. The use of "cognitive ability" is confusing. It would be helpful if the authors could be more specific about when they are talking about DA scores, when they are talking about other measures of cognitive ability.

Authors' response: This is now been clarified within the text of the revised manuscript.

Reviewer(s)' Comments to Author:10. It is really helpful when the authors explain in the text what low or high confidence scores mean. Please do this everywhere! For example p15 line 29.

Authors' response: We have now added this explanation whenever this reference is made, to help guide the reader, as suggested.

Minor comments:

Reviewer(s)' Comments to Author:p4 line 12: "accurately estimate self-confidence" should be "accurately estimate ability".

Authors' response: This has been corrected.

Reviewer(s)' Comments to Author:p4 line 49: I think "self-efficiency" should be "self efficacy".

Authors' response: This has now been corrected.

Reviewer(s)' Comments to Author:p5. line 43 to 56 I found it really difficult to understand the description of the DA test.

Authors' response: We have now reworded this description to increase its readability and for clarification.

Reviewer(s)' Comments to Author:p6. line 26. Remove comma after "Previous work".

Authors' response: This has been removed.

Reviewer(s)' Comments to Author:p7 line 37. What does "online confidence as an ability" mean? Confidence in ability to answer DA questions correctly? I think the authors mean confidence on the DA test vs general confidence. I suspect many readers won't be familiar with the concept of psychological traits, so this needs a bit more explaining.

Authors' response: We agree this could have been clearer. We have therefore expanded this section a little, in order to explain the difference between the conceptualisation of 'confidence as an ability' and 'confidence as a trait'.

Reviewer(s)' Comments to Author:p7 line 41. "lift and scatter" needs more explaining here. Also why lift and not elevate?

Authors' response: We agree that the term 'elevation and scatter' would be more appropriate, and indeed is the term used by the original proponents of the approach, and we have now made this change in the manuscript. We have also changed the wording to explain what we mean by this concept. Also, the reference to the method has now been moved to the methods section, which appears more appropriate.

Reviewer(s)' Comments to Author:p8 line 9. "their own judgement (adjusted for 'lift and scatter') " feels like it's missing a few words. Judgement of what? Confidence?

Apologies- this could have been clearer. We have now clarified the sentence to make it clear that we are talking about a candidate's judgement regarding their own ability at the decision analysis items. In addition rephrased as 'elevation and scatter'.

Reviewer(s)' Comments to Author:p9. line 29. preceding year is confusing. They take the test in the calendar year they apply.

This is correct. This sentence has now been reworded accordingly.

Reviewer(s)' Comments to Author:p9 line 31 The sentence starting "All data were placed in a safehaven..." is repeated later on. It also needs a bit more detail. Was this done within UKMED? Which safehaven? Who did the linkage and the cleaning?

We have expanded on this, stating that the raw data were placed in the Health Informatics Centre Safe Haven by UKCAT, and were subsequently cleaned, managed, linked and analysed by the research team inside the Safe Haven. Specifically, the safe haven used is the same one that hosts the UKMED, though these data were separate to UKMED.

Reviewer(s)' Comments to Author:p11. How many medical schools were UKCAT schools at the time? I also think it's possible that BMAT schools are different. This is mentioned in the Discussion but a little more could be made of it.

At the time the study there were roughly 30 medical schools that were part of the UKCAT consortium, and 35 UK medical schools, though the number varied during the study period with Aston and UCLAN launching in 2015.. We agree that it is possible that medical schools using the BMAT differed in some way, and we have made mention of this in the discussion section, as suggested. However, as we have pointed out in our discussion, it is our experience of working with medical selection assessments

is that almost all candidates who sit the BMAT also sit the UKCAT, though the converse is not true. This is presumably the case because medical school applicants would generally not want their options at application limited to the small number of universities that use the BMAT as their selection assessment.

Reviewer(s)' Comments to Author:p11. There are two steps between getting an offer and starting at medical school: accepting an offer and meeting that offer. This could be briefly explained before going on to "of those who entered medical school..." (line 55).

This has now been rephrased in that section of the manuscript, to make it clear that 'success' in this case is more than receiving an offer. The section now appears to read more clearly/

VERSION 2 – REVIEW

REVIEWER	Rob Meijer University of Groningen
REVIEW RETURNED	27-Nov-2019
GENERAL COMMENTS	thanks for the changes, I have no further comments